# Effectiveness of Intramammary Antibiotics, Internal Teat Sealants, or Both at Dry-Off in Dairy Cows: Clinical Mastitis and Culling Outcomes

**DOI:** 10.3390/antibiotics11070954

**Published:** 2022-07-15

**Authors:** Sharif S. Aly, Emmanuel Okello, Wagdy R. ElAshmawy, Deniece R. Williams, Randall J. Anderson, Paul Rossitto, Karen Tonooka, Kathy Glenn, Betsy Karle, Terry W. Lehenbauer

**Affiliations:** 1Veterinary Medicine Teaching and Research Center, School of Veterinary Medicine, University of California, Davis, Tulare, CA 93277, USA; eokello@ucdavis.edu (E.O.); welashmawy@ucdavis.edu (W.R.E.); dvmwilliams@ucdavis.edu (D.R.W.); paulrossitto799@gmail.com (P.R.); khtonooka@ucdavis.edu (K.T.); mycoqueen@yahoo.com (K.G.); tlehenbauer@vmtrc.ucdavis.edu (T.W.L.); 2Department of Population Health and Reproduction, School of Veterinary Medicine, University of California, Davis, Tulare, CA 95616, USA; 3Department of Internal Medicine and Infectious Diseases, Faculty of Veterinary Medicine, Cairo University, Giza 12211, Egypt; 4California Department of Food and Agriculture, Animal Health Branch, Sacramento, CA 95814, USA; randy.anderson@cdfa.ca.gov; 5Cooperative Extension, Division of Agriculture and Natural Resources, University of California, Orland, CA 95963, USA; bmkarle@ucanr.edu

**Keywords:** dry cow therapy, mastitis, culling, subclinical mastitis, teat end score, udder hygiene, California mastitis test

## Abstract

Intramammary antibiotic (AB) and internal teat sealants (TS) infusion at dry-off have been used to prevent intramammary infections (IMI) in dairy cows during the dry period and reduce the risk of mastitis during the dry period and subsequent lactation. A randomized clinal trial was completed on eight California dairy herds to estimate the effects of different dry cow therapies (AB, TS, AB + TS or None) on clinical mastitis and culling. A total of 1273 cows were randomized to one of the four treatment groups over summer and winter seasons. For each enrolled cow, microbiological testing was done on quarter milk samples collected from the first detection of clinical mastitis within the first 150 days in milk (DIM) in the subsequent lactation. Statistical analysis was done using generalized linear mixed models. There were no significant differences in the odds of clinical mastitis or culling between cows treated with AB, TS, or AB + TS compared to the controls. Dry cow therapy with AB and/or TS had no statistically significant effect on clinical mastitis and cow culling during the first 150 DIM.

## 1. Introduction

Mastitis continues to be a major challenge to the dairy industry, and multiple efforts have been implemented to reduce its impact on dairy cattle health and welfare [1,2]. Dry cow therapy (DCT), a common practice among dairy farmers worldwide, is the intramammary infusion of long-acting antibiotics (AB) at the end of the lactation to control intramammary infections (IMI) at dry-off and reduce the risk of mastitis during the dry period and the subsequent lactation [3]. The decision to treat all 4 quarters of each cow at dry-off with intramammary antibiotics is commonly referred to as Blanket Dry Cow Treatment (BDCT). Alternatively, antibiotic treatment of specific cows, or individual quarters within a cow that satisfy specific criteria, is known as Selective Dry Cow Treatment (SDCT), and has been evaluated in various studies [4,5,6]. In the United States (US), BDCT is a common practice, with approximately 93% of dairy cows treated with intramammary antibiotics at dry-off representing approximately 80.3% of the US dairy herds [7].

Due to the growing concern for antimicrobial resistance, regulations have been implemented nationally and internationally to reduce unnecessary antibiotic use [8]. Previous studies compared the effects of using BDCT and SDCT on the health outcomes of dairy cows (e.g., clinical mastitis, culling) with no significant differences or additional benefits except that BDCT resulted in increased antibiotic usage [4,5]. A recent study found that the performance of culture-guided SDCT and algorithm-guided SDCT on risks of clinical mastitis and culling were similar to BDCT; however, the study was not conducted throughout the year, thereby missing any seasonal effects on mastitis [9]. Huxley et al. (2002) reported no significant differences in the number of IMI caused by major pathogens after calving between cows that received internal teat sealants and those that received long-acting AB at dry-off. In 2014, a Canadian study revealed no significant differences in the risk of post-calving IMI and clinical mastitis within 120 days in milk (DIM) in the subsequent lactation between SDCT based on farm culture and BDCT [5]. A different study in New York between 2016–2017 revealed no significant differences in new IMI infection risk, culling or clinical mastitis between high-risk cows that received BDCT and low-risk cows (classified according to a culture based on a farm algorithm) randomized to receive internal teat sealants (TS), or AB and TS (AB + TS) [6]. Hence, current research suggests that not all cows will benefit from intramammary antibiotic infusions at dry-off which may advance antibiotic stewardship on dairies by limiting dry-off treatment to cows at risk of negative health and production outcomes. However, the aforementioned studies restricted their herds to low-bulk tank somatic cell counts (BTSCC) at <250,000 cells/mL or assessed SDCT on low-mastitis-risk cows with either somatic cell counts (SCC) <200,000 cells/mL or ≤one clinical mastitis in the lactation prior to dry-off. As a result, prior research findings may not apply to all herds, especially those that differ substantially from these criteria.

Use of a BTSCC cut-off for implementation of SDCT on dairy herds assumes that all cows in such herds will benefit from antibiotics; however, this is not the case, given that the BTSCC is an average of the milking herd’s SCC which can be heavily biased by outlier cows. Furthermore, SDCT based on an SCC obtained from monthly tests at the cow or quarter level assumes a similar status at dry-off, which depends on the duration elapsed and new or cured intramammary infections since testing. Finally, controlled trials can be used to validate SDCT algorithms by providing baseline estimates for negative health and production outcomes and their costs—data that could be used for algorithm validation and cost-effectiveness.

Our research aimed to fill gaps in the current knowledge on the efficacy of dry cow therapy across herds with a natural range of BTSCC, seasons, and antibiotics treatments available to producers via a controlled experimental design, with the overall goal of generating the data required to develop and validate a SDCT algorithm. The objectives of this research were to capture a wide spectrum of herd management styles and common practices to estimate the effects of dry cow therapies (AB, internal teat sealants (TS) or both (AB + TS)) on the health outcomes of dairy cows—specifically clinical mastitis and culling during the dry period and the first 150 DIM in the subsequent lactation.

## 2. Results

### 2.1. Herd Characteristics

The mean number of lactating cows in the enrolled herds was 1782 (SE ± 347) with a range of 750 to 3689 milking cows. Herd breeds included in the study were: Holstein only, 50% (4 herds); Holstein, Jersey and crossbred, 25% (2 herds); Holstein and Jersey, 12.5% (one herd); and Jersey and crossbred, 12.5% (one herd). All enrolled herds used Dairy Comp305 (Valley Ag Software, Tulare, CA, USA) as dairy herd record-keeping systems. The dry-off schedule for cows at the end of lactation was once a week in all enrolled herds except for one herd, which dried cows every two weeks. The BTSCC in the month prior to enrollment on the two San Joaquin County herds was <200,000 and between 300,000 and 400,000 cells/mL, respectively; in the Stanislaus County herd, it was <200,000 cells/mL; in the three Tulare County study herds, it was <200,000, between 200,000 and 300,000, and between 300,000 and 400,000 cells/mL, respectively; and in the two Kings County herds, it was between 200,000 and 300,000 cells/mL, respectively. The mean of the last 6 months BTSCC 305,571 (SE ± 41,508) was reported for only 7 of the 8 enrolled herds.

### 2.2. Mastitis Control

Mastitis control questions focused on record keeping, vaccination against mastitis, dry cow therapy, culling and milking practices. There were no outbreaks of contagious mastitis in the enrolled herds, while one herd reported isolation of Staphylococcus aureus from a single bulk tank sample and another herd reported isolation of both Staphylococcus aureus and Mycoplasma bovis from bulk tank samples in the six months prior to the study. Seven herds recorded clinical mastitis events using Dairy Comp 305 (Valley Ag Software, Tulare, CA, USA), and one herd used paper records. All enrolled herds vaccinated their cows against coliform mastitis according to label directions using commercially available vaccines with gram-negative core antigens.

All enrolled herds used DCT to prevent and control mastitis during the dry period and subsequent lactation. Only four herds used internal teat sealant at dry-off (Orbeseal^®®^, Zoetis, Kalamazoo, MI, USA), however, cows allocated to receive internal teat sealants on the remaining study herds were provided the same product. Five herds practiced BDCT at dry-off while the other three herds practiced SDCT. Herds practicing SDCT selected cows to be treated based on either history of clinical mastitis during the current lactation (2 herds) or SCC >200,000 cell/mL for the last two test results (1 herd).

The milking schedule for the enrolled herds was either two times/day (4 herds) or three times/day (3 herds); cows from one herd were milked three or two times/day in the high and low lactating pens, respectively. Only five of the enrolled herds had a wash pen to wash the cows’ udders prior to milking followed by a drip pen to allow cows’ udders to dry prior to milking for approximately 20 min (SE ± 4.47). Pre-milking teat dip was practiced on only seven herds using either teat cups (3 herds) or teat sprayers (4 herds) with chlorine-based disinfectant (3 herds), iodine-based disinfectant (2 herds) or peroxide (2 herds). All enrolled herds applied post-milking teat dip using chlorine-based disinfectant (4 herds), iodine-based disinfectant (3 herds) or lactic acid (1 herd).

### 2.3. Management Practices

#### 2.3.1. Dry Cow Management

Six of the enrolled dairies housed dry cows on the same premises as the lactating herd, while the remaining two kept dry cows offsite on separate premises before transporting them back to the dairy post-calving. Dry cows kept on the same premises were housed in either dry lot pens (5 herds) or freestall pens (1 herd). Freestall pens were bedded with dried manure on a weekly basis while dry lot pens were bedded under the shades with either dried manure (3 herds) or almond shells (2 herds). Only four herds out of the six that kept their dry cows onsite used flush for cleaning of lanes in the dry cow pens either once/day (1 herd) or three times/day (3 herds). Pen designs for close-up cows of the six herds that kept their dry cows onsite were dry lot (4 herds), freestall (1 herd) and bedded pack (1 herd), with three herds bedding with dried manure, two herds with almond shells and one herd with straw. To facilitate drying of the bedding surface in close up pens, two herds incorporated crushed limestone and two herds incorporated bedding raking. Only four herds had a flush alley in the close-up pens and flushed either three times/day (3 herds) or twice/day (1 herd). Out of the six herds that housed their dry cows onsite, five of them had separate maternity pens and one herd had no maternity pen.

#### 2.3.2. Lactating Cow Management

Lactating pens of the enrolled herds were: freestall only (4 herds), freestall plus dry lot pens (3 herds) and bedded pack (1 herd). Lactating cow pens were bedded with dried manure (5 herds), almond shell (1 herd), recycled sand (1 herd) and dried manure plus woodchips (1 herd). Bedding refill of lactating cow pens was weekly (4 herds), twice/week (1 herd), every 10 days (1 herd) and as needed (2 herds). All lactating pens had flush lanes with flushing frequency of six times/day (2 herds), four times/day (1 herd), three times/day (4 herds) and two times/day (1 herd). In addition, two herds also manually removed manure from lactating cow pens.

### 2.4. Enrollment and Baseline Comparisons

Frequency of cow enrollment occurred based on herd dry-off management. Specifically, this was weekly on all the study herds, with the exception of 1 herd, where it occurred every two weeks until the study sample size was achieved. A total of 1273 cows were presented for enrollment over two seasons (538 cows in winter, 735 cows in summer). Cows that did not meet the enrollment criteria (*n* = 140) were excluded (Figure 1).

A total of 1133 cows were enrolled in the study; 27 were excluded due to errors transcribing their identification, resulting in a total of 1106 cows included for final analysis. Total number of cows enrolled by herd were: 109 and 278 from the two San Joaquin County herds, 55 from the Stanislaus County herd; 91, 107, and 173 from the three Tulare County herds, and 86 and 207 from the two Kings County herds. Over the study period, a total of 480 cows were enrolled in the winter and 653 in the summer cohorts. After random assignment, the distribution of the enrolled cows by treatment group (276 cows received AB; 253 cows received TS; 282 cows received AB + TS; and 295 cows received no therapy) had no significant differences (*p* > 0.05) between the treatment groups’ mean days in milk at enrollment, daily milk production (Kg), DHI test-day SCC linear score prior to enrollment or dry period length following enrollment (Table 1).

In addition, there were no significant differences (*p* > 0.05) in parity, breed, season of enrollment or occurrence of mastitis during the enrollment lactation (Table 2). Of the 1106 cows, 45 cows were culled during the dry period and another 166 cows were culled during the first 150 days post-calving (89 of the 166 were culled between calving and the first Dairy Herd Improvement Association (DHIA), test).

### 2.5. Clinical Mastitis during the First 150 DIM

Less than a quarter of the cows in each treatment group developed clinical mastitis during the first 150 DIM in the lactation after enrollment, with only numerical differences among different treatment groups (AB = 21.80%, TS = 21.90%, AB + TS = 17.77% and None = 22.96%). There was a significant difference in the proportion of clinical mastitis among parities, 24.71% of cows with parity ≥ 3 developed clinical mastitis, in comparison to 16.02% in second lactation cows (*p* < 0.01). Similarly, 45.28% cows with two or more teats with a score of four had clinical mastitis, in comparison to 19.64% in cows with lower teat end scores (*p* < 0.01), and 26.94% of cows with California Mastitis Test (CMT) ≥2 in any quarter had clinical mastitis, compared to 19.11% in cows with CMT score <2 in any quarter (*p* < 0.01). Table 3 summarizes the GLMM for the occurrence of clinical mastitis in the first 150 DIM of the lactation after enrollment. The random intercept for dairy had a variance estimate of 0.722 with 95% confidence intervals of 0.246 to 2.119. Additional analyses showed no statistically significant differences in odds of clinical mastitis by type of AB administered at dry-off.

There were no differences in odds of clinical mastitis during the first 150 DIM among the treated groups in comparison to the control group. In addition, multiple comparisons showed no differences in odds of clinical mastitis between different treatment contrasts. Crossbred cows had significantly lower odds for clinical mastitis during the 150 DIM compared to Holstein cows. In contrast, Jersey cows had only numerically lower odds for clinical mastitis in comparison to Holstein cows. Parity, teat end score and CMT score were significantly associated with clinical mastitis. Specifically, cows of a third or greater parity at dry-off (compared to second parity), cows with two or more teats with teat end score of four at dry-off (compared to none or 1 teat) and cows that had CMT score ≥ two for one or more quarters at dry-off (compared to CMT score < 2 in all quarters) had higher odds for clinical mastitis during the first 150 DIM.

### 2.6. Cow Culling

Results of the generalized linear mixed model predicting cow culling during the dry period after enrollment are presented in Table 4. The random intercept for dairy had a variance estimate of 0.078, with 95% confidence intervals of 0.003 to 2.411. There were no statistical differences among therapies with respect to the proportions of cows culled during the dry period following enrollment (AB = 3.6%, TS = 4.3%, AB + TS = 4.3%, None = 4.1%; *p* > 0.05). Multiple comparisons showed no differences in odds of culling during the dry period between different treatment contrasts. History of clinical mastitis during the enrollment lactation and teat end score ≥3 were the statistically significant predictors for cow culling during the dry period (*p* < 0.05). Cows that had mastitis during the enrollment lactation had greater odds of culling in comparison to cows that did not have mastitis (*p* < 0.01). Cows with one or more teats with a teat end score of three or more had greater odds of culling during the dry period in comparison with cows with teat end scores less than three (*p* = 0.03).

Logistic regression results predicting cow culling during the first 150 DIM in the subsequent lactation are summarized in Table 5. There were numerical differences among therapies with respect to culling during the first 150 days in the lactation after enrollment (AB = 13.2%, TS = 16.5%, AB + TS = 14.4%, None = 18.4%, *p* > 0.05). Multiple comparisons showed no differences in odds of culling in the first 150 DIM in the remaining treatment contrasts. Breed, parity, number of mastitis events during the enrollment lactation and season of enrollment were significant predictors (*p* < 0.01) for cow culling in the first 150 DIM of lactation after enrollment. Crossbred cows had significantly lower odds of culling between calving and 150 DIM compared to Holsteins. Similarly, cows in their third or greater lactation (compared to second lactation) and cows with history of one or more clinical mastitis events at the enrollment lactation (compared to those with none) were at significantly greater odds of culling. Cows enrolled in the summer had greater odds of culling in their first 150 DIM compared to those enrolled in the winter. There were no differences among the treatment groups with respect to culling during the entire study period (dry-off to 150 DIM). Table 6 summarizes the logistic regression model for culling during the entire study period including breed, parity, number of clinical mastitis events during the enrollment lactation and enrollment season as significant predictors (*p* < 0.01).

Additional analyses comparing the type of AB administered at dry-off showed no statistically significant differences in odds of culling during dry period, first 150 DIM post-calving, or the entire period from dry-off to 150 DIM post-calving.

## 3. Discussion

The current randomized clinical trial investigated associations between different dry cow therapies, including AB, TS, and AB + TS, on dairy cows’ clinical mastitis and culling outcomes. Dry cow therapy (AB, TS, AB + TS) only numerically reduced clinical mastitis and culling during the first 150 DIM compared to untreated cows. In contrast, breed, parity, teat end score and CMT were significantly associated with clinical mastitis in the first 150 DIM. Culling during the dry cow period was only associated with history of clinical mastitis prior to dry-off and teat end score at dry-off. In addition to breed and parity, history of clinical mastitis prior to dry-off, and dry-off season were significantly associated with culling during the first 150 DIM.

### 3.1. Clinical Mastitis during the First 150 DIM

Our study showed that there were no significant differences in odds of clinical mastitis post-calving between the trial groups, which was in agreement with previous studies despite differences in study design [10,11]. Others have reported significant differences in post-calving clinical mastitis between cows treated at dry-off with AB + TS compared to AB only, which could be attributed to differences in herds and their management practices, or to the statistical methods used [12,13,14].

The current study showed that crossbreeds had the lowest odds of developing clinical mastitis, in comparison to pure breeds (Holstein and Jersey), which could be due to better udder conformation, enhanced immune function, and genetic variations [15]. Similarly, previous studies have reported differences among cow breeds in terms of susceptibility to clinical mastitis [16,17].

Our study showed that cows of third lactation or greater had significantly higher odds of clinical mastitis after calving in comparison to second lactation cows, which could be related to their higher risk of exposure to mastitis organisms in the previous lactations and higher productivity, which may predispose them to new IMI [18,19]. Previous studies reported higher odds of clinical mastitis with increasing parity [18,20].

The current study showed that cows with two or more teats with a teat end score of four showed higher odds of developing clinical mastitis compared to cows with teat end scores of less than four, which was in agreement with previous studies [21,22]. The teat sphincter is the first physical barrier against bacterial infection of the mammary tissue, and cows with smooth teat ends have greater ability to form keratin plugs, which has a bactericidal effect [23]. In contrast, teats with rough ends may lead to incomplete closure of the teat orifice, which hinders formation of a keratin plug and can act as a scaffold that increases the chances for bacterial entry, multiplication and risk of mastitis [24,25]. The latter finding confirmed previous research, which showed that increasing teat end callosity, thickness, and roughness were associated with higher risk of IMI [22]. Moreover, other studies found that increasing teat callosity was influenced by parity, which may help to explain the observed relationship between these factors [21,26].

The California mastitis test is a rapid field diagnostic for subclinical mastitis. Our study showed that cows with a CMT score of three or more in any quarter had higher odds of developing clinical mastitis, perhaps due to the presence of subclinical infections at dry-off that persisted and resulted in clinical mastitis post-calving [27,28].

### 3.2. Cow Culling

Our study showed no statistical differences between treated groups (AB, TS, AB + TS) in comparison to the non-treated group in culling, either during the dry period or during the first 150 DIM after calving. Similar findings were reported in previous studies [4,6]. Cows with a history of two or more events of clinical mastitis prior to dry-off had the highest odds of culling, which could be associated with = higher risk of developing clinical mastitis, lower productivity, or no response to mastitis treatment in the subsequent lactation [4]. Similarly, cows with one or more teats with teat end scores of three or greater at dry-off had higher odds of culling, which could be associated with a higher risk for developing clinical mastitis, as teats with rough ends are usually associated with failure of complete closure of the teat orifice [24].

The lowest odds of cow culling were reported among crossbreeds, in comparison to pure breeds (Holstein and Jersey), which could be due to differences in productivity, reproduction, and genetics [29]. Higher odds of culling were reported among cows of higher parity (≥3), which could be attributed to higher risk of developing other disease conditions (mastitis, ketosis, hypocalcemia, lameness) as age increases. Similarly, previous studies also showed that the culling risk increased with increasing parity [29,30]. Our models showed a significant seasonal difference in culling, with higher culling during the summer season, perhaps due to poor reproduction and low productivity associated with heat stress [31].

### 3.3. Limitations

A limitation of the current study was use of the herd veterinarian-prescribed antibiotic treatment for cows allocated AB or AB + TS. This was by design, to increase the external validity of findings. However, this may have introduced additional variability in the effects of these treatment groups compared to use of a single antibiotic. Dairies enrolled in the study were from the top four milk-producing counties in California, where the management practices, weather conditions and cow housing may differ from other countries or regions. Similar studies may be needed in other locations to address other climatic, seasonal, and housing conditions. Cow breeds enrolled in the study were the common milk-producing breeds in California (Holstein, Jersey, and crossbreeds), while in other countries, the milk breeds may differ. The enrolled herds did not record clinical mastitis during the dry period; hence, future studies that explore novel monitoring systems of cows in the dry pen may be needed to study the association between different dry cow therapies and mastitis during the dry period.

## 4. Material and Methods

### 4.1. Study Design

A single-blinded, controlled, randomized clinical trial was conducted on 8 dairies with cows enrolled between December 2016 and March 2017 (winter season) and again between June 2017 and September 2017 (summer season), with all 8 herds sampled in both seasons (IACUC protocol number 19761). The trial sample size was calculated based on the difference in post-calving IMI prevalence between treatments, an outcome that will be the subject of a second manuscript, currently in preparation. Briefly, the sample size assumed a 15.3% prevalence of IMI with BDCT [5] and an odds ratio (OR) for a treatment effect between the study groups (1:1 enrollment ratio) ranging from 0.1 to 0.6 in increments of 0.1. The computation showed that milk samples from 1200 cows (composite samples) would provide a minimum of 82% power at a 5% alpha error rate for detecting significant differences between groups.

### 4.2. Herd Selection

Eight herds were selected from across the Northern San Joaquin Valley (NSJV) and the greater Southern California (GSCA) regions [32]. Specifically, 2 of the study herds were from San Joaquin, 1 from Stanislaus, 3 from Tulare, and 2 from Kings Counties. Inclusion criteria for the study herds included; (1) enrollment in DHIA testing for at least a year prior to the study; (2) willingness to allow study personnel access to the herds’ DHIA software records; (3) willingness to allow access to mastitis samples collected by farm personnel for the study cows, either through the herd veterinarian or the milk quality lab that routinely received the herd’s milk samples for bacteriological culture; and (4) willingness to comply with the trial guidelines and protocol. The eight herds enrolled were a convenience sample, with BTSCC ranging from low to high in order to capture the combined effects of herd, cow and management factors associated with milk quality. No restrictions were made on the use of TS or mastitis vaccines, but each enrolled herd could not have any history of contagious mastitis pathogen outbreaks (*Mycoplasma* spp., *Strep. agalactiae* and *Staph. aureus*) in the previous six months.

### 4.3. Enrollment Survey

During each herd’s enrollment and at the end of the study, a questionnaire was completed with the help of the dairy owner or manager to capture herd demographics, management, milking, and dry-off practices. Specifically, the survey captured information about the herd size and breed distribution, rolling herd average milk production and the most recent and highest BTSCC in the last six months. In addition, questions regarding the herds’ management practices inquired about dry-off frequency (per month), type of dairy record software used and for how long, whether the herd belonged to DHIA and the frequency of milk component testing. Adult cow housing questions in this section inquired about the dairy pen types, dimensions, use of lime, bedding materials, the frequency of pen or bed raking and/or refilling, and flushing as applicable. In addition, data regarding the prevalence of mastitis and common mastitis pathogens detected in the herd and the use of mastitis vaccines were collected. The survey also included questions about milking practices, such as the number of times the herd was milked each day and if it varied by parity or pen, the use of a wash pen and/or drip pen prior to milking, length of time cows were in the drip pen if used, type of pre- and/or post-milking teat dip used, use of a separate pen for fresh cows other than the hospital pen, whether the milkers wore gloves and how often and under what circumstances (e.g., after certain number of milkings or if they tear) they were replaced. The remaining questions were directed to dry-off practices, specifically the use of BDCT or SDCT, dry-off antibiotic used, if any internal or external TS were used, modifications in feed or times cows were milked prior to dry-off.

### 4.4. Cow Enrollment

Cows presented for dry-off on the study herds were enrolled on a weekly basis except for one herd where enrollment occurred every two weeks based on their dry-off schedule. In addition, enrollment continued on the study herds for 1 to 4 weeks each season based on herd size until the study sample size was achieved. On dry-off day, only cows with no ongoing health events were enrolled; this was verified by normal udder and milk, lack of health codes on the cow’s record and no record of treatments in the past two weeks prior to enrollment. In addition, only cows with body condition score >2.5, no lameness, and four functional quarters were enrolled. Cows with clinical mastitis [abnormal udder (swollen, hot, tender), teats (teat injuries) or milk (abnormal color, consistency)] at dry-off were excluded from the trial and treated with the dairy’s standard dry cow treatment protocol. Enrolled cows were randomly assigned to receive either the dairy’s standard dry cow intramammary AB, TS, both (AB + TS) or no treatment (None). Cows were followed from dry-off to 150 DIM in the subsequent lactation with data on clinical mastitis and culling collected.

### 4.5. Dry-Off Sampling (S1) and Therapy

At dry-off, cows without health conditions or antibiotic treatments in the previous two weeks were identified from the dry-off list and observed at the parlor upon entry for low body condition score (<2.5) and lameness. In addition to recording cow ID and breed, each cow’s teat end score (1) normal, smooth teat end and no ring; (2) smooth teat end with ring; (3) rough teat end with ring; (4) very rough cracked teat end with ring) and udder hygiene score (scale of 4 with 1 being clean and 4 being dirty) were recorded [33]. Next, each cow’s teat ends were cleaned after the pre-milking teat dip, and the first 2–3 strips of milk were discarded from each quarter, followed by testing with the California Mastitis Test using a scale of five points (CMT: Negative, the mixture remained liquid, no evidence of precipitations; Trace, the mixture had slight precipitation; 1, a clear precipitate formed with no tendency for gel formation; 2, the mixture thickened with some gel formed; 3, the mixture thickened with complete gel formation and stuck to the bottom of the paddle) [28]. Scores for udder hygiene, teat end abnormities and CMT were recorded by study personnel without knowledge of treatment allocation. Enrolled cows were randomly allocated to one of the treatment groups (AB, TS, AB + TS, or None) using a prepared standard random number list generated in Microsoft Excel (Excel 2016, Microsoft Corp., Redmond, WA, USA). After random allocations were verified, each cow’s quarter was treated with either AB, TS, AB followed by TS or none. Cows allocated to a TS received Orbeseal (Zoetis, Kalamazoo, MI, USA), while cows allocated to an AB treatment received the same antibiotic dry-off treatment prescribed by their herd veterinarian. Specifically, the dry-off AB prescribed by the study herd veterinarians were: Spectramast DC (Zoetis, Kalamazoo, MI, USA) on one Kings County herd; Dry-Clox (Boehringer Ingelheim Vetmedica Inc., St. Joseph, MO, USA) on one Kings County herd and the Stanislaus County herd; ToMorrow (Boehringer Ingelheim Vetmedica Inc., St. Joseph, MO, USA) on two San Joaquin County herds and one Tulare County herd; and Quartermaster (Zoetis, Kalamazoo, MI, USA) on 2 Tulare County herds. Finally, each teat was dipped in the herd’s routine post-milking teat dip solution and leg bands were attached to both hindlimbs of the enrolled cows for ease of identification during follow up.

### 4.6. Follow-Up of Enrolled Cows

Enrolled cows were followed during weekly herd visits for up to 150 DIM in the subsequent lactation. All cow events, including clinical mastitis and culling (sale or death), were identified from the study dairies DHIA record backups (Dairy Comp305, Valley Ag Software, Tulare, CA, USA).

### 4.7. Statistical Analyses

A relational database (Access, Microsoft Corp., Redmond, WA, USA) was created to house and manage data enrollment, bacterial culture results and DHIA records for the study herds. Data analyses were conducted using Stata (Stata Corp. 2017. Stata Statistical Software: Release 15. Stata Corp LLC, College Station, TX, USA). Baseline comparisons and differences between treatment groups were conducted using ANOVA for continuous variables and Chi Square test of independence for categorical variables.

The dichotomous outcomes denoting the presence or absence of clinical mastitis (CM) during the first 150 days post calving, and culling. For culling, three models were explored predicting culling during the dry period, during the first 150 DIM of the subsequent lactation, or both (dry-off to 150 DIM).

The explanatory variables considered were: treatment (AB, TS, AB + TS, None), season (fall to winter or spring to summer), the cow level variables’ breed (Holstein, Jersey, Mixed), parity, teat end score (at enrollment), CMT result (at enrollment), udder hygiene score (at enrollment), occurrences of mastitis during enrollment lactation and any prior lactations, days in milk at dry-off (at enrollment), and SCC (at enrollment). Cow parity at enrollment was categorized into first, second and third or greater lactation. However, post-calving parity was categorized into second and third or greater lactations. Teat end score was explored with different categorizations including two categories (Yes, No) for each score level (scores 2–4) and the number of teats at each score level. Similarly, CMT score was explored with different categorizations including two categories (Yes, No) for each score level (scores Trace-3) and the number of quarters at each score level. Udder hygiene score was explored as a variable with four levels and categorized into two levels, with scores 1 and 2 as one level and scores 3 and 4 as the second level. History of clinical mastitis was explored during the enrollment lactation and lactations prior to the enrollment lactation as dichotomous variables (Yes, No) and ordinal variables specifying the number of clinical mastitis events recorded. Days in milk at time of enrollment was also explored in our models. Somatic cell counts for the last 6 months milk production prior to enrollment were explored as dichotomous variables (Yes, No) for different cut-offs, starting at 200,000 cells/mL up to 500,000 cells/mL in increments of 50,000 cells/mL for each test, starting from the most recent test and working backwards. In addition, the last test day SCC prior to enrollment was explored as a continuous variable.

#### 4.7.1. Modeling Clinical Mastitis during the First 150 DIM

A generalized linear mixed model (GLMM) with a logit link was used to model the logit of the probability of clinical mastitis at the cow level on the study dairies for the different treatment groups (AB, TS, AB + TS, None). Cows with multiple clinical mastitis events were considered only once. The model-estimated odds ratio (OR), a measure of the association between the predictors and clinical mastitis, ranged from 0 to infinity with an OR of 1 interpreted as no association, <1 for variables protective against mastitis, or >1 for variables that are risk factors. Significant associations were identified at the 5% level of significance. Equation (1) summarizes the GLMM used to model the occurrence of clinical mastitis during the first 150 DIM in the subsequent lactation. Dairy was included in the model as a random effect variable $u_{i},\;i=\left\{1,2,3,4,5,7,8\right\}$; $\beta_{0}$ = intercept, and βX = fixed effects for the explanatory variables. Univariate models for each fixed effect variable, with dairy as a random effect, were specified to explore their associations with the outcome.
(1)$$\mathrm{Logit} [P\left({\mathrm{CM}}_{i}\right)]=\beta_{0}+\beta X+u_{i}$$

In the case where a random effect’s variance was inestimable, a logistic regression model was specified with cluster-robust standard errors. Model building followed a manual backward approach with treatment forced into the models, confounding by known confounders were assessed using the method of change in estimates and effect modifiers (interaction) assessed using significance testing [34]. Selection of the final model was based on the Akaike information criterion (AIC) to select between competing models, with lower values denoting better model goodness of fit [35]. Bonferroni-adjusted multiple comparisons were performed between treatment groups.

#### 4.7.2. Modeling Cow Culling

Three models were used to model the occurrence of cow culling: (1) culling during the dry period, (2) culling during 150 DIM in the lactation following enrollment and (3) culling during both the dry period and during 150 DIM. Logit link GLMMs, similar to Equation (1) but with the outcome being one of each of the three culling outcomes, were specified to estimate the odds of culling at the cow level on the study dairies, as explained by the different treatment groups (AB, TS, AB + TS, None). Logistic regression was specified with cluster-robust standard errors if random effects were inestimable. The final models for each of the culling outcomes were developed by following the same procedures described for the clinical mastitis model.

## 5. Conclusions

Certain dairy cows may benefit from dry cow therapy depending on their risk for negative outcomes, such as higher parity cows with higher odds of clinical mastitis and culling during the first 150 DIM or cows with higher teat end or CMT scores that have higher odds of developing clinical mastitis during the first 150 DIM. Cows with a history of clinical mastitis in the enrollment lactation had higher odds of culling in the following lactation. Crossbreeds had a significant reduction in clinical mastitis and culling during 150 DIM in the subsequent lactation, in comparison to Holsteins and Jerseys.

## Figures and Tables

**Figure 1 antibiotics-11-00954-f001:**
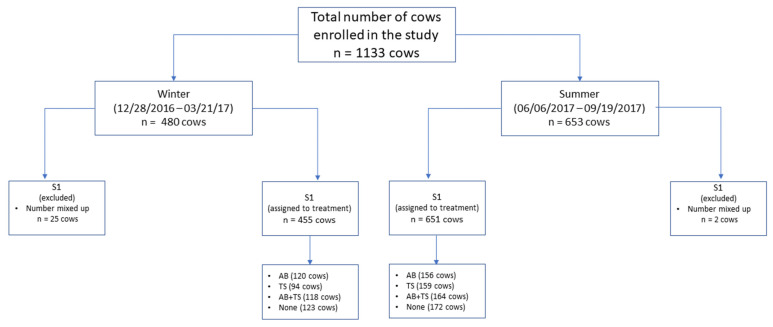
Schematic representation of the enrolled cows in the randomized field trial for dry cow therapy. AB (cows received only intramammary antibiotic tubes at dry-off), TS (cows received only internal teat sealants at dry-off), AB + TS (cows received both intramammary antibiotic tubes and internal teat sealants at dry-off) and None (cows did not receive any treatments at dry-off).

**Table 1 antibiotics-11-00954-t001:** Comparison of the continuous variable baseline traits for the enrolled dairy cows (*n* = 1106 cows) from 8 herds after allocation to the four treatment groups (AB, TS, AB + TS, None) ^1^. There were no significant differences between the treatment groups (*p* > 0.05).

Variable	Treatment																			
	Antibiotic (AB) Only (*n* = 276)					Teat Sealant (TS) Only (*n* = 253)					Both (AB + TS) (*n* = 282)					None (*n* = 295)				
	*n*	Mean	SE	95% CI		*n*	Mean	SE	95% CI		*n*	Mean	SE	95% CI		*n*	Mean	SE	95% CI	
DIM at enrollment	276	332.0	4.32	324.0	341.0	253	331.0	3.62	324.0	339.0	281	331.0	3.76	324.0	339.0	295	328.0	3.45	322.0	335.0
Dry period	266	61.0	1.15	59.0	64.0	242	60.0	0.84	59.0	620.0	270	59.0	0.84	58.0	61.0	283	59.0	0.79	58.0	61.0
Daily milk production (kg) ^2^	274	26.0	0.60	24.8	27.2	253	25.6	0.54	24.5	26.6	280	26.1	0.55	25.0	27.2	294	26.1	0.54	25.0	27.2
Linear Score	274	2.8	0.12	2.6	3.1	253	2.9	0.12	2.7	3.2	280	3.0	0.11	2.2	3.2	294	3.1	0.11	2.9	3.3

^1^ AB (cows received only intramammary antibiotic tubes at dry-off), TS (cows received only internal teat sealants at dry-off), AB + TS (cows received both intramammary antibiotic tubes and internal teat sealants at dry-off) and None (cows did not receive any treatments at dry-off). ^2^ Milk (kg): daily milk production recorded by the last Dairy Herds Improvement Association (DHIA) test prior to enrollment.

**Table 2 antibiotics-11-00954-t002:** Comparison of the categorical variable baseline traits for the enrolled dairy cows (*n* = 1106 cows) from 8 herds after allocation to the four treatment groups (AB, TS, AB + TS, None) ^1^.

Variable	Levels	Percent of Cows Enrolled by Treatment				Total	*p*-Value
		AB (*n* = 276)	TS (*n* = 253)	AB + TS (*n* = 282)	None (*n* = 295)		
Parity at enrollment	First	119/475 (25.1%)	102/475 (21.5%)	128/475 (27.0%)	126/475 (26.5%)	475	0.65
	Second	81/334 (24.2%)	85/334 (25. 5%)	74/334 (22.2%)	94/475 (28.1%)	334	
	≥Third	76/297 (25.6%)	66/297 (22.2%)	80/297 (26.9%)	75/297 (25.3%)	297	
Mastitis during enrollment lactation	Yes	17/88 (19.3%)	18/88 (20.5%)	22/88 (25.0%)	31/88 (35.2%)	88	0.25
	No	259/1018 (25.4%)	235/1018 (23.1%)	260/1018 (25.5%)	264/1018 (25.9%)	1018	
Breed	Holstein	185/723 (25.6%)	167/723 (23.1%)	177/723 (24.5%)	194/723 (26.8%)	723	0.94
	Jersey	50/204 (24.5%)	46/204 (22.6%)	53/204 (26.0%)	55/204 (27.0%)	204	
	Cross	41/179 (22.9%)	40/179 (22.4%)	52/179 (29.1%)	46/179 (25.7%)	179	
Season	Winter	120/455 (26.4%)	94/455 (20.7%)	118/455 (25.9%)	123/455 (27.0%)	455	0.49
	Summer	156/651 (24.0%)	159/651 (24.4%)	164/651 (25.2%)	172/651 (26.4%)	651	

^1^ AB (cows received only intramammary antibiotic tubes at dry-off), TS (cows received only internal teat sealants at dry-off), AB + TS (cows received both intramammary antibiotic tubes and internal teat sealants at dry-off) and None (cows did not receive any treatments at dry-off).

**Table 3 antibiotics-11-00954-t003:** Generalized linear mixed model with logit link predicting clinical mastitis from calving to ≤150 days in milk among different treatment groups (AB, TS, AB + TS, None) ^1^ on California dairies (*n* = 8 herds).

Variable	Level	Cows (% Mastitis, Number Enrolled at Risk of Clinical Mastitis)	Odds Ratio	Standard Error	*p*-Value	95% Confidence Limits	
						Lower	Upper
Treatment	None	(22.96%, 283)	Referent				
	AB	(21.80%, 266)	0.87	0.196	0.56	0.56	1.36
	TS	(21.90%, 242)	0.97	0.223	0.92	0.62	1.52
	AB + TS	(17.77%, 270)	0.72	0.167	0.17	0.46	1.14
Breed	Holstein	(23.03%, 686)	Referent				
	Jersey	(22.11%, 199)	0.61	0.198	0.12	0.32	1.15
	Cross	(12.50%, 176)	0.37	0.124	<0.01	0.19	0.72
Parity	Second	(16.02%, 462)	Referent				
	≥3	(24.70%, 599)	1.65	0.300	<0.01	1.15	2.35
Number of teats with teat end score = 4 at enrollment	<2	(19.64%, 1008)	Referent				
	≥2	(45.28%, 53)	2.67	0.896	<0.01	1.38	5.15
CMT ≥ 2 at any quarter at enrollment	No	(19.11%, 816)	Referent				
	Yes	(26.94%, 245)	1.54	0.296	0.02	1.06	2.25
Intercept			0.24	0.090	<0.01	0.11	0.50

^1^ AB (cows received only intramammary antibiotic tubes at dry-off), TS (cows received only internal teat sealants at dry-off), AB + TS (cows received both intramammary antibiotic tubes and internal teat sealants at dry-off) and None (cows did not receive any treatments at dry-off).

**Table 4 antibiotics-11-00954-t004:** Generalized linear mixed model with logit link predicting culling during dry period among different treatment groups (AB, TS, AB + TS, None) ^1^ on California dairies (*n* = 8 herds).

Variable	Level	Cows (% Culled, Number Enrolled at Risk of Culling)	Odds Ratio	Standard Error	*p*-Value	95% Confidence Limits	
						Lower	Upper
Treatment	None	(4.06%, 295)	Referent				
	AB	(3.62%, 276)	0.93	0.414	0.88	0.39	2.22
	TS	(4.34%, 253)	1.17	0.508	0.70	0.50	2.74
	AB + TS	(4.25%, 282)	1.13	0.477	0.77	0.49	2.58
Mastitis during the enrollment lactation	No	(3.53%, 1018)	Referent				
	Yes	(10.22%, 88)	2.94	1.178	<0.01	1.34	6.45
Teat end score of one or more teats at enrollment	<3	(3.07%, 813)	Referent				
	≥3	(6.82%, 293)	2.03	0.687	0.03	1.04	3.94
Intercept			0.02	0.09	<0.01	0.01	0.05

^1^ AB (cows received only intramammary antibiotic tubes at dry-off), TS (cows received only internal teat sealants at dry-off), AB + TS (cows received both intramammary antibiotic tubes and internal teat sealants at dry-off) and None (cows did not receive any treatments at dry-off).

**Table 5 antibiotics-11-00954-t005:** Logistic regression model predicting culling after calving up to 150 days in milk among different treatment groups (AB, TS, AB + TS, None) ^1^ on California dairies (*n* = 8 herds).

Variable	Level	Cows (% Culled, Number Enrolled at Risk of Culling)	Odds Ratio	Standard Error	*p*-Value	95% Confidence Limits	
						Lower	Upper
Treatment	None	(18.37%, 283)	Referent				
	AB	(13.15%, 266)	0.69	0.170	0.13	0.43	1.12
	TS	(16.52%, 242)	0.88	0.211	0.59	0.55	1.41
	AB + TS	(14.44%, 270)	0.79	0.189	0.33	0.50	1.26
Breed	Holstein	(17.78%, 686)	Referent				
	Jersey	(15.07%, 199)	0.85	0.197	0.50	0.54	1.34
	Cross	(7.95%, 176)	0.43	0.126	<0.01	0.24	0.76
Parity ^2^	Second	(9.95%, 462)	Referent				
	≥3	(20.03%, 599)	1.88	0.362	<0.01	1.29	2.74
Clinical mastitis events in enrollment lactation	0	(14.05%, 982)	Referent				
	1	(31.74%, 63)	2.65	0.776	<0.01	1.49	4.70
	≥2	(50.00%,16)	5.11	2.662	<0.01	1.838	14.184
Season	Winter	(11.38%, 439)	Referent				
	Summer	(18.64%, 622)	1.63	0.303	<0.01	1.13	2.35
Intercept			0.11	0.026	<0.01	0.07	0.18

^1^ AB (cows received only intramammary antibiotic tubes at dry-off), TS (cows received only internal teat sealants at dry-off), AB + TS (cows received both intramammary antibiotic tubes and internal teat sealants at dry-off) and None (cows did not receive any treatments at dry-off). ^2^ Parity in this table has categories second and ≥3 since all enrolled cows at dry-off were parous females.

**Table 6 antibiotics-11-00954-t006:** Logistic regression model predicting total culling (dry period + after calving until 150 DIM) among different treatment groups (AB, TS, AB + TS, None) ^1^ on California dairies (*n* = 8 herds).

Variable	Level	Cows (% Culled, Number Enrolled at Risk of Culling)	Odds Ratio	Standard Error	*p*-Value	95% Confidence Limits	
						Lower	Upper
Treatment	None	(21.69%, 295)	Referent				
	AB	(16.30%, 276)	0.73	0.164	0.16	0.47	1.13
	TS	(20.15%, 253)	0.91	0.201	0.68	0.59	1.40
	AB + TS	(18.08%, 282)	0.84	0.183	0.42	0.55	1.29
Breed	Holstein	(21.72%, 723)	Referent				
	Jersey	(17.15%, 204)	0.76	0.62	0.20	0.50	1.15
	Cross	(9.49%, 179)	0.40	0.110	<0.01	0.23	0.69
Parity ^2^	First	(12.42%, 475)	Referent				
	Second	(21.55%, 334)	1.67	0.33	0.01	1.13	2.47
	≥3	(26.93%, 297)	2.06	0.412	<0.01	1.39	3.05
Clinical mastitis events in enrollment lactation	0	(17.09%, 1018)	Referent				
	1	(38.57%, 70)	2.88	0.763	<0.01	1.71	4.84
	≥2	(55.55%, 18)	4.98	2.481	<0.01	1.88	13.22
Season	Winter	(14.50%, 455)	Referent				
	Summer	(22.27%, 651)	1.55	0.260	<0.01	1.11	2.15
Intercept			0.15	0.031	<0.01	0.10	0.22

^1^ AB (cows received only intramammary antibiotic tubes at dry-off), TS (cows received only internal teat sealants at dry-off), AB + TS (cows received both intramammary antibiotic tubes and internal teat sealants at dry-off) and None (cows did not receive any treatments at dry-off). ^2^ Parity in this table has 3 categories (first, second, ≥3) to account for culling of the first lactation heifers that were culled during the dry period.

## Data Availability

The data presented in this study are available upon reasonable request from the corresponding author. The data are not publicly available, as the study-associated dairy owners did not consent to publishing it alongside the article.

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
