# Peer review of "Effectiveness of Intramammary Antibiotics, Internal Teat Sealants, or Both at Dry-Off in Dairy Cows: Clinical Mastitis and Culling Outcomes"

_antibiotics, 2022, doi:10.3390/antibiotics11070954_

Round 1
Reviewer 1 Report
I have enjoyed reading your article on the effectiveness of intramammary antibiotics and teat seal-ants on clinical mastitis and culling outcomes. The manuscript is well-written and authors did an excellent job with data analysis. This reviewer, on the other hand, has major and minor comments that need to be addressed before accepting the manuscript for publishing.
Major comments:
- Authors assigned cows in four groups, one of them AB group where antibiotics prescribed by herds veterinarian. The analysis did not account for the variations between the antibiotics used in AB group on the clinical mastitis and culling outcomes. Even though I appreciate how tough it is to regulate this variation, the author can at least study the extent of it within the AB group. Within the AB group, authors may perform statistical test to evaluate the association between outcomes and antibiotic type administered.
- The result section should be reorganized. Sections 1 to 3 have a lot of information about the farm management which could be summarized in 1 or 2 supplementary tables. The authors intend to provide an overview about the management practices in the herds involved in the study however this reviewer believes it is unrelated to the manuscript's major objectives.
- Authors included herds as random effect in the models. Report the total variation in the model that may be attributed to the herds.
- Line 377-382: sample size calculation, is this calculation assumes that quarters within a cow are independent? and how cows assigned to each group?
Minor comments
Line 23: (AB, TS, AB + TS) abbreviations should be written in full name.
Line 23: delete “dairy cow health, specifically” to be “on clinical mastitis ….”
Line 58: change “long-acting antibiotics” to “long-acting AB”
Line 100-105: explain the reasons for including the mean BTSCC of 7 out 8 herds only?
Line 117-121: these information (AB used in each herd) were already listed under M&M lines 454 to 459. Delete repeated information.
Line 165-174: This is confusing!! .. 1273 cow presented and only 1133 enrolled. Is this means 140 cows did not meet the enrollment criteria? .. Out of the 1133 cows enrolled in the study, 27 cows were excluded due to error? .. Please make it clear!
Line 197: Table 2, change “Treatment” to “% of cows enrolled in treatment” then deleted “%” in the 2nd row.
Table 3 to 5: add the herd random effect.
Line 307: authors reported that the difference could be related the various antibiotics used, even in the study there is no control for variations that may resulted from using different antibiotic in each herd.
Line 349: “lower immunity”.. This is incorrect as adult cows have well developed immune system in comparison with calves. Authors may attribute it to the higher chance of exposure to a source of infection with age increase!
Line 358-359: It would be fascinating to see how much diversity there was as a result of choosing various AB within group AB and how that affected the results.
Line 378: change the reference to journal style.
Line 426: delete “and”
Line 467: statistical analysis, it would be excellent if authors included a table describing the outcomes and explanatory variables with their levels, so that readers can easily follow and understand the analysis undertaken.
Author Response
Thank you, please see attached responses.

Reviewer 2 Report
The aim of this clinical trial was to evaluate the efficacy of 4 different dry-off treatment approaches to reduce clinical mastitis and culling during the following lactation in dairy cows; in detail, intramammary antibiotic infusion (AB), internal teat sealant (TS), and AB+TS were compared to the controls (no treatment).
Results showed no significant differences among treatments for the health outcome variables; moreover, the association between parity, CMT and/or teat-end score with clinical mastitis and culling was observed.
The manuscript is well written and adequately detailed; however, some revisions are needed before its acceptance for publication.
Minor revisions
Line 185: The total number of the AB+TS reported for treatment group must be edited (n.282 instead of 281)
Line 193: The results reported in this section refer to table 1. Author must edit the reference.
Major revision
Results section: Categorical variables should be summarized with absolute frequency (percentage) (e.g., 10/100 (10%)) in order to standardize the format for an easier interpretation of the results.
Table 2: Authors must provide a brief description of the results showed in the Table 2, although no statistical differences were observed.
Line 294-297: This assumption is not evidence-based and it is not statistically supported by the findings of the study. Authors should remove or edit this sentence to avoid any possible misunderstandings.
Author Response
Please see attached responses.

Round 2
Reviewer 2 Report
The Authors fully accepted the revision proposed and edited the manuscript as suggested.
In my opinion, the paper is now suitable for the publication in the journal.